# Comparative Analysis of Genetic Alterations, HPV-Status, and PD-L1 Expression in Neuroendocrine Carcinomas of the Cervix

**DOI:** 10.3390/cancers13061215

**Published:** 2021-03-10

**Authors:** Daisuke Takayanagi, Sou Hirose, Ikumi Kuno, Yuka Asami, Naoya Murakami, Maiko Matsuda, Yoko Shimada, Kuniko Sunami, Masaaki Komatsu, Ryuji Hamamoto, Mayumi Kobayashi Kato, Koji Matsumoto, Takashi Kohno, Tomoyasu Kato, Kouya Shiraishi, Hiroshi Yoshida

**Affiliations:** 1Division of Genome Biology, National Cancer Center Research Institute, 5-1-1 Tsukiji, Chuo-ku, Tokyo 104-0045, Japan; dtakayan@ncc.go.jp (D.T.); kuno-ik@mc.pref.osaka.jp (I.K.); yuasami@ncc.go.jp (Y.A.); maimatsu@ncc.go.jp (M.M.); yoshimad@ncc.go.jp (Y.S.); ksunami@ncc.go.jp (K.S.); tkkohno@ncc.go.jp (T.K.); 2Department of Obstetrics and Gynecology, The Jikei University School of Medicine, 3-19-18, Nishishinbashi, Minato-ku, Tokyo 105-8471, Japan; s-hirose@jikei.ac.jp; 3Medical Oncology, Osaka International Cancer Institute, 3-1-69, Otemae, Chuo-ku, Osaka 541-8567, Japan; 4Department of Obstetrics and Gynecology, Showa University School of Medicine, 1-5-8 Hatanodai, Shinagawa-ku, Tokyo 142-8666, Japan; matsumok@mui.biglobe.ne.jp; 5Department of Radiation Oncology, National Cancer Center Hospital, 5-1-1 Tsukiji, Chuo-ku, Tokyo 104-0045, Japan; namuraka@ncc.go.jp; 6Division of Molecular Modification and Cancer Biology, National Cancer Center Research Institute, 5-1-1 Tsukiji, Chuo-ku, Tokyo 104-0045, Japan; maskomat@ncc.go.jp (M.K.); rhamamot@ncc.go.jp (R.H.); 7Cancer Translational Research Team, RIKEN Center for Advanced Intelligence Project, 1-4-1 Nihonbashi, Chuo-ku, Tokyo 103-0027, Japan; 8Department of Gynecology, National Cancer Center Hospital, 5-1-1 Tsukiji, Chuo-ku, Tokyo 104-0045, Japan; maykobay@ncc.go.jp (M.K.K.); tokato@ncc.go.jp (T.K.); 9Department of Diagnostic Pathology, National Cancer Center Hospital, 5-1-1 Tsukiji, Chuo-ku, Tokyo 104-0045, Japan

**Keywords:** neuroendocrine carcinomas, cervical cancer, next-generation sequencing, targeted therapy, HPV, PD-L1

## Abstract

**Simple Summary:**

Patients with neuroendocrine carcinoma of the cervix (NECC) have limited treatment options due to its rarity and aggressiveness. In this study, we performed a comparative genetic analysis between 25 NECC and other cervical cancer types (180 squamous cell carcinoma, 53 adenocarcinoma, and 14 adenosquamous carcinoma). Furthermore, the expression of programmed cell death-ligand 1 (PD-L1) was assessed by immunohistochemistry. *PIK3CA* and *TP53* were commonly altered genes in cervical cancer, while *SMAD4*, *RET*, *EGFR*, and *APC* were NECC-specific altered genes. Of note, 11 NECC cases showed at least one actionable mutation linked to molecular targeted therapies, and 14 cases showed more than one combined positive score for PD-L1 expression. These results may boost the generation of effective treatment strategies for NECC in the future.

**Abstract:**

Neuroendocrine carcinoma of the cervix (NECC) is a rare and highly aggressive tumor with no efficient treatment. We examined genetic features of NECC and identified potential therapeutic targets. A total of 272 patients with cervical cancer (25 NECC, 180 squamous cell carcinoma, 53 adenocarcinoma, and 14 adenosquamous carcinoma) were enrolled. Somatic hotspot mutations in 50 cancer-related genes were detected using the Ion AmpliSeq Cancer Hotspot Panel v2. Human papillomavirus (HPV)-positivity was examined by polymerase chain reaction (PCR)-based testing and in situ hybridization assays. Programmed cell death-ligand 1 (PD-L1) expression was examined using immunohistochemistry. Somatic mutation data for 320 cases of cervical cancer from the Project GENIE database were also analyzed. NECC showed similar (*PIK3CA*, 32%; *TP53*, 24%) and distinct (*SMAD4*, 20%; *RET*, 16%; *EGFR*, 12%; *APC*, 12%) alterations compared with other histological types. The GENIE cohort had similar profiles and *RB1* mutations in 27.6% of NECC cases. Eleven (44%) cases had at least one actionable mutation linked to molecular targeted therapies and 14 (56%) cases showed more than one combined positive score for PD-L1 expression. HPV-positivity was observed in all NECC cases with a predominance of HPV-18. We report specific gene mutation profiles for NECC, which can provide a basis for the development of novel therapeutic strategies.

## 1. Introduction

Neuroendocrine carcinoma of the cervix (NECC) is an uncommon and highly aggressive tumor. Based on Surveillance, Epidemiology, and End Results (SEER) data from the National Cancer Institute, the annual incidence of this cancer is 0.05/100,000 [1]. In Japan, NECC reportedly accounts for less than 2% of all cervical cancers [2]. Approximately 50% of patients with NECC have distant metastases at the time of diagnosis, and the median survival is 18.2 months for all stages [1]. The standard systemic therapy for NECC has not been established in a randomized prospective clinical trial owing to its rare occurrence. Some retrospective studies suggest that chemotherapy complies with that for small-cell lung cancer (SCLC) [3,4] or more common cervical cancer types, such as squamous cell carcinoma (SCC). Platinum-based chemotherapy, combined with etoposide, is generally used as an adjuvant or first-line chemotherapy [3,5], whereas regimens including topotecan, paclitaxel, and bevacizumab are used for some recurrent cases of NECC [4]. Nevertheless, the prognosis of patients with NECC remains dismal; thus, novel and more efficient systemic therapeutic options, including molecular targeted therapy or immunotherapy, are required.

Genomic data of NECC are expected to provide basis for novel and more efficient treatment options for this aggressive tumor. In fact, molecular targeted therapies have contributed to the improvement of patient outcomes in various cancers [6]. However, few studies have focused on molecular targeted therapies for NECC [7,8]. Unfortunately, genomic data available for NECC are still very limited, particularly for Asian patients, although some previous studies reported recurrent genetic alterations involving the phosphoinositide 3-kinase (PI3K)/protein kinase B (AKT)/mechanistic target of rapamycin (mTOR), mitogen-activated protein kinase (MAPK), and P53 pathways in NECC [9,10,11,12]. One of these reports on comparison of genomic profiles between NECC and SCLC demonstrated considerably different molecular characteristics [11], whereas another report revealed genetic similarities between NECC and other major cervical cancer subtypes, including SCC and adenocarcinoma (ADC), using published genomic data [12]. Considering the genomic differences between NECC and SCLC, the present treatment strategy for NECC complying with that for SCLC could be optimized based on the specific genetic features of NECC. Furthermore, deciphering genetic differences between NECC and other subtypes of cervical cancer is expected to help identify specific targets for the treatment of NECC.

The relationship between genotypes of human papillomavirus (HPV) and mutational profiles of NECC remains to be elucidated. High-risk HPV is reportedly associated with most cervical cancers, including SCC, ADC, and adenosquamous carcinoma (ASC) of the cervix [13]. Recently, NECC was reported to be associated with high-risk HPV, primarily HPV16 and HPV18 [14]. Although we previously reported the association between HPV genotypes and histological types and genetic alterations in cervical cancer [15], we could not demonstrate features specific to NECC owing to a very limited number of such cases.

In this study, we aimed to decipher the genetic characteristics of NECC compared with cervical cancer of other histological types using targeted sequencing and analysis of data available in a public database. Furthermore, we explored the links between the identified mutations and targeted therapies. We also elucidated the HPV genotypes in NECC and their association with genetic alterations.

## 2. Materials and Methods

### 2.1. Patients

The study protocol was approved by the Institutional Review Board of the National Cancer Center Research Institute (2017–136) and the study was conducted in accordance with the ethical guidelines of the Helsinki Declaration. Written informed consent was obtained from all patients through an opt-out form. Patients who refused to provide consent were excluded from the study.

Two-hundred and eighty-seven patients with pathologically confirmed cervical cancers, including NECC (n = 26), SCC (n = 191), ADC (n = 55), and ASC (n = 15), who received treatment between 2002 and 2018 at the National Cancer Center Hospital, Japan were retrospectively enrolled. Of the 287 cases, sequencing data from 272 patients met the quality control criteria and were included in this study. All the cases were reviewed by at least two gynecological pathologists, and the pathological diagnoses were confirmed according to the World Health Organization (WHO) tumor classification [16].

### 2.2. DNA Preparation and Next-Generation Sequencing

Genomic DNA was extracted from formalin-fixed paraffin-embedded (FFPE) tumor tissues using the QIAamp DNA FFPE Tissue Kit (Qiagen, Hilden, Germany), according to the manufacturer’s instructions. Library construction was performed using purified genomic DNA (50 ng) obtained from the tumor tissues and the Ion AmpliSeq^TM^ Cancer Hotspot Panel v2 (Thermo Fisher Scientific, Waltham, MA, USA), which targets approximately 2800 Catalog of Somatic Mutations in Cancer (COSMIC) mutational hotspot regions of 50 cancer-related genes. An Ion AmpliSeq^TM^ Custom Panel that was designed for the *TP53* gene (coverage: all coding regions) using Ion AmpliSeq^TM^ Designer (https://www.ampliseq.com, accessed on 8 November 2019) was also used. Sequencing was performed using the Ion Proton platform (Thermo Fisher Scientific). For quality control, samples with a mean read depth of coverage over 1000 and a base quality score of 20 (≤1% probability of being incorrect) were selected, which accounted for 90% of the total reads.

### 2.3. Classification of Oncogenic/Pathogenic Mutations

The sequencing reads were mapped to the University of California, Santa Cruz (UCSC) human reference genome GRCh37, and data analysis was carried out using the Torrent Suite Software v5.0.4 (Thermo Fisher Scientific). Somatic mutations were initially selected using the following criteria: (1) variant allele frequency of somatic mutations was >4% in tumor tissues, (2) single nucleotide polymorphisms were removed if they showed a threshold allele frequency ≥0.01 in either the National Heart, Lung, And Blood Institute (NHLBI) Grand Opportunity Exome Sequencing Project (ESP6500; http://evs.gs.washington.edu/EVS/ (accessed on 1 July 2019)) or the integrative Japanese Genome Variation Database (iJGVD, 20181105; https://ijgvd.megabank.tohoku.ac.jp/ (accessed on 1 July 2019)), and (3) the mutations were registered as “pathogenic/likely pathogenic variants” in the ClinVar or as “oncogenic/likely oncogenic variants” in OncoKB (http://oncokb.org) databases using oncokb-annotator, commit 8910b65 (accessed on 29 June 2019). All the selected variants were then evaluated manually using the Integrative Genomics Viewer (IGV; http://www.broadinstitute.org/igv/ (accessed on 29 May 2019)).

### 2.4. Detection of Copy Number Alterations Using the Taqman Assay

Copy number alterations were detected by real-time genomic polymerase chain reaction (PCR) using the TaqMan copy number assay and the ABI 7900HT real-time PCR system (Applied Biosystems). Four genes (*PIK3CA*, *ERBB2*, *PTEN*, and *STK11*) were selected from among the 50 targeted genes in the Ion AmpliSeq™ Cancer Hotspot Panel v2, and the frequency of copy number alterations in these four genes was detected to be >5% in the TCGA dataset of cervical cancer patients within cBioPortal. All TaqMan probes, including *PIK3CA* (ID Hs02202946_cn), *ERBB2* (ID Hs02803918_cn), *PTEN* (ID Hs05128032_cn), *STK11* (ID Hs04013006_cn), and *RNase P* (cat. no. 4403328), which was used as a reference, were purchased from Thermo Fisher Scientific. Genome data were analyzed using the ABI PRISM 7900HT Sequence Detection Software CopyCaller v2.1 (Thermo Fisher Scientific) for copy number analysis. Copy number amplification was defined as the process resulting in the presence of >8 copies, whereas copy number loss was defined as the process resulting in the presence of <1.2 copy.

### 2.5. Identification of Human Papillomavirus (HPV) Genotyping by Sanger Sequencing

HPV genotypes were identified as follows: Genomic DNA (10 ng) was amplified using PCR for two distinct HPV genomic regions. The E6/E7 region of HPV was amplified using the primer set pU-1M/pU2R (HPVpU-1M: 5′-TGTCAAAAACCGTTGTGTCC-3′ and HPVpU-2R: 5′-GAGCTGTCGCTTAATTGCTC-3′); the region containing the HPV L1 gene was amplified using the primer set GP5+/GP6+ (GP5+: 5′-TTTGTTACTGTGGTAGATACTAC-3′, GP6+: 5′-GAAAAATAAACTGTAAATCATATTC-3′). PCR reactions were performed using the TaKaRa PCR Human Papillomavirus Typing Set (TaKaRa Bio Inc., Shiga, Japan). PCR products were purified using the NucleoSpin Gel (TaKaRa Bio Inc.) or a PCR Clean-up Kit (TaKaRa Bio Inc.). Sanger sequencing was performed using an ABI 3130xl DNA Sequencer (Applied Biosystems, Foster City, CA, USA), according to the manufacturer’s instructions. Similarity between the obtained sequences and various HPV genotypes in the GenBank database was determined by Basic Local Alignment Search Tool (BLAST) analysis (https://blast.ncbi.nlm.nih.gov/Blast.cgi (accessed on 17 June 2019)).

### 2.6. Detection of High-Risk HPV Types

To determine the frequency of HPV-positive samples, we performed in situ hybridization (HPV-ISH) using HPV-III High Risk probes (Roche Diagnostics, Mannheim, Germany), according to the manufacturer’s instructions. The HPV-ISH assay is able to detect high-risk HPV genotypes in cervical cancer specimens, including HPV-16, HPV-18, HPV-31, HPV-33, HPV-35, HPV-45, HPV-52, HPV-56, HPV-58, and HPV-66.

### 2.7. Immunohistochemistry

Immunohistochemistry for retinoblastoma protein, RB1, was performed on 4 μm thick paraffin-embedded tissue sections, which had the same tumor areas provided for genetic analysis, using an RB1-specific mouse monoclonal antibody (clone G3-245, 1:400, BD Pharmingen). Tumors were scored as negative for RB1 protein if more than 95% of cells showed no RB1 staining, with positive nuclear staining in the surrounding endothelial cells serving as an internal control. The expression of programmed cell death-ligand 1 (PD-L1) was examined in tumor cells and infiltrating histiocytes by immunohistochemistry using a rabbit polyclonal antibody (SP142). The expression of PD-L1 was scored by both tumor proportion score (TPS) and combined positive score (CPS), as previously described [17].

### 2.8. Large-Scale Genomic Datasets

Somatic mutations called from targeted sequencing data in the American Association for Cancer Research (AACR) Project Genomics Evidence Neoplasia information Exchange (GENIE) database (GENIE version 8.0-public) were downloaded as MAF files via the Synapse Platform (http://www.synapse.org/genie (accessed on 2 July 2020)).

### 2.9. Clinical Association and Actionability Analysis

OncoKB, a precision oncology knowledge base containing information regarding the actionability and therapeutic implications of specific genomic alterations in cancer patients, was used. Somatic mutations were classified into four levels. Gene aberrations with evidence levels of 1–3B were identified as actionable mutations for molecular targeted drugs.

### 2.10. Statistical Analysis

Statistical analysis was performed using the SPSS (version 27.0, IBM, Armonk, NY, USA) and R (version 3.6.0; R Foundation, Vienna, Austria) software. Categorical variables were compared using the chi-square test or Fisher’s exact test for small sample sizes. Continuous variables were compared using the unpaired Student’s *t*-test. All *p*-values were two-tailed, and *p*-values < 0.05, were considered statistically significant. Overall survival (OS) was defined as the time from the start of treatment to death from any cause. Survival curves were computed using Kaplan–Meier estimates with log-rank tests. Cox regression analysis was applied to test the predictors of OS and to calculate hazard ratios with 95% confidence intervals (95% CI).

## 3. Results

### 3.1. Patient Demographics

The clinicopathological characteristics of 272 patients are summarized in Table 1. The median age of patients with NECC was 43 years (range, 28–68 years), which was lower than that of patients with other histological types. The rate of occurrence of advanced International Federation of Gynecology and Obstetrics (FIGO) stages (III/IV) in NECC was 32%, which was higher than that of ADC and ASC (18.9% and 7.1%, respectively). A total of 249 patients (91.5%) were positive for high-risk HPV; these included 25 patients with NECC (100%), 169 with SCC (93.8%), 44 with ADC (83.0%), and 11 with ASC (78.6%).

### 3.2. Different Genetic Alterations between Neuroendocrine Carcinoma of the Cervix (NECC) and Other Histological Types in Patients with Japanese Cervical Cancer

The profiles of the frequent genetic alterations in the NECC and other histological types are shown in Figure 1. We identified 64 mutations in patients with NECC as being oncogenic/likely oncogenic or pathogenic/likely pathogenic mutations in the OncoKB and ClinVar databases, respectively. In patients with NECC, the mutations included 50 non-synonymous mutations, 13 stop-gain mutations, and one splicing-site mutation (Appendix A). For the most frequently mutated genes in patients with NECC, *PIK3CA* and *TP53* variants were detected in 6/25 (24%) of the patients each, followed by *SMAD4* variants in 5/25 (20%), *PTEN* variants in 4/25 (16%), and *RET* variants in 4/25 (16%) patients. Copy number losses were detected in 4/25 (16%) patients for *STK11* and in 3/25 (12%) patients for *PTEN*. Copy number amplifications were detected in 3/25 (12%) patients for *PIK3CA*. The PI3K pathway is frequently activated in cervical cancer, and the PI3K pathway-related gene *PIK3CA* was frequently altered in all histological types (Figure 1 and Figure 2A). The frequency of mutation of *TP53* in patients with NECC was significantly higher than that in those with SCC (24.0% vs. 6.1%, *p* < 0.05, Figure 2A). The frequency of mutations in *RET*, *EGFR*, *SMAD4*, *PTPN11*, and *APC* in patients with NECC were higher than those in patients with other histological types. The mTOR signaling-related gene, *STK11*, previously reported by our group as a poor prognostic factor for cervical cancer [15], was altered significantly more frequently in patients with NECC than in those with SCC (16.0% vs. 4.4%, *p* < 0.05, Figure 2A). No statistically significant differences were observed between the frequency of MAPK pathway-related genes, *KRAS* and *BRAF*, and *NRAS* mutations in patients with NECC than in those with other histological subtypes.

### 3.3. Comparison of Frequency of Mutation between NECC and Other Histological Types of Cervical Cancer in the Project GENIE Database

We also examined differences in the mutation frequency between NECC and the other histological subtypes of cervical cancer according to the Project GENIE data (Appendix A), and compared the results with those obtained for our cohort. The Project GENIE database contained 11 NECC, 223 SCC, 50 ADC, and 36 ASC cases (Appendix A). The median age of patients with NECC was 38 years (range, 29–51 years), which was lower than those of patients with other histological types. The data were obtained from patients mainly from the USA, and only 24 cases (7.5%) were from Asia. Mutations in *PIK3CA* were most frequently observed and detected in all the histological types (Figure 2B). The frequency of *TP53* mutations in patients with NECC was higher than that in patients with other histological types, and was significantly higher than that in patients with SCC (27.2% vs. 6.7%, *p* < 0.05, Figure 2B). In addition, genetic mutations in *RB1* were detected more frequently in patients with NECCs than in those with SCC and ADC (27.2% vs. 5.8%, 0%, *p* < 0.05; Figure 2B).

### 3.4. Immunohistichemical Detection of the Expression of RB1 and PD-L1

Because the sequence targeted by us only covered the hotspot regions of *RB1* mutations, we could not detect the relatively infrequent mutations of *RB1*. Therefore, we additionally assessed the expression of RB1 by immunohistochemistry in all the 25 patients with NECC. Loss of RB1 expression was not observed in any of the samples.

For identifying the putative therapeutic targets for NECC, we also evaluated the expression of PD-L1, which is reportedly a predictor of response to immunotherapy, by immunohistochemical staining. Of the 25 patients with NECC, none of the cases showed more than 1% TPS, whereas 56% (14/25) of the cases presented more than 1 CPS (Appendix A).

### 3.5. Actionable Mutations in NECC

Actionable mutations registered as evidence levels of 1–3B in OncoKB were detected in 11 (44%) patients with NECC (Figure 3). Accordingly, 8 (32%) and 4 (16%) patients with NECC may have benefited from mTOR/AKT/PI3K and RET inhibitors, respectively. *RET* mutation is a therapeutic target not found in other histological types.

### 3.6. Association between HPV Genotypes and Genetic Alterations in NECC

Approximately 80% of the patients with NECC had detectable HPV-18, and HPV-18-positivity was statistically more frequent in patients with NECC than in those with other histological subtypes (Figure 4). Next, we examined the association between HPV genotypes and genetic alterations frequently detected in NECC; however, there was no significant difference in mutation profiles between NECC with and without HPV-18 (Appendix A).

### 3.7. Prognostic Factors of NECC

Two patients with NECC were excluded from the prognostic analysis because follow-up data were not available. For others, the median survival time was 29.1 months (range, 0.53–140 months). The OS was significantly worse among patients with advanced FIGO stages (median OS of stage I: 129.3 vs. stage II: 39.9 vs. stage III: 49.4 vs. stage IV: 18.5 months; log-rank, *p* = 0.002; Appendix A). Patients with *KRAS* mutations had significantly shorter OS than those without these mutations (log-rank *p* < 0.001; Appendix A).

In the univariate Cox regression analysis, the FIGO stage and *KRAS* mutation were the variables that were significantly associated with shorter OS. The multivariate Cox survival analysis indicated that the FIGO stage and *KRAS* mutation remained significant prognostic factors for shorter OS (HR: 5.95, 95% CI: 1.58–22.34, *p* = 0.0083 and HR: 12.92, 95% CI: 1.12–148.89, *p* = 0.04; Appendix A).

## 4. Discussion

NECC is an extremely aggressive malignancy with high mortality rates, even in patients diagnosed at an early stage [3]. Systemic treatment strategies based on prospective clinical trials are lacking owing to the low incidence of NECC. Novel therapeutics, including molecular-targeted therapy or immunotherapy, have been in demand. In the present study, we analyzed the genetic features of NECC and the clinical value of genetic alterations associated with molecular targeted therapies. We unravel the genetic similarities and differences between NECC and other histological types of cervical cancer. We also show that more than half of the patients with NECC may benefit from molecular targeted therapies and immune checkpoint inhibitors. Furthermore, an important role of high-risk HPV infections in patients with NECC has been demonstrated.

Our targeted sequencing results, comparing NECC with other subtypes of cervical cancers, reveal some similarities. Alterations in *PIK3CA* were more frequently detected among all histological types, predominantly at two sites (E542K and R545K) in the helical domain of *PIK3CA*. In Project GENIE, *PIK3CA* mutations in patients with NECC were less frequent (9.1%), whereas those in patients with other histological types were comparable to our results. Activation of the PI3K pathway through *PIK3CA* regulates various transformed phenotypes as well as the growth and differentiation of HPV-16 and HPV-18-immortalized cells; thus, it may play a pivotal role in HPV-induced carcinogenesis [18]. Our cohort and the Project GENIE data showed a similar distribution of *TP53* mutations, more frequently observed in patients with NECC than in those with SCC and being similar to that in patients with ADC. Moreover, mutations in the MAPK pathway genes, including *KRAS*, *BRAF*, and *NRAS*, were observed at similar frequencies among the histological types. This observation was comparable to that reported previously [12]. Of note, these similarities in genetic alterations between NECC and the other histological types suggest that targeted therapies for genetic alterations, which are widely observed in cervical cancer, can be extended to NECC.

Notably, we identified that several recurrent gene mutations, such as those in *EGFR*, *RET*, *SMAD4*, *APC*, *CTNNB1*, and *PTPN11*, are limited in NECC. These gene mutations are uncommon in cervical cancer [19]. *EGFR* regulates multiple intracellular target pathways and affects a wide range of biological processes [20]. *EGFR* mutations are significantly correlated with a highly differentiated grade in patients with cervical cancer [21]. *SMAD4*, which regulates the canonical transforming growth factor (TGF)-β signaling pathway in porcine granulosa cells, was mutated in 5 (20%) patients with NECC. *SMAD4* plays a critical role in tumor progression and induces proinflammatory cytokines, such as interleukin (IL)-5, IL-6, and IL-13, which might induce a tumor-promoting microenvironment [22]. Oncogenic *RET* mutations were first identified in 4 (16%) patients with NECC in this study. To date, only two cervical SCC cases harboring *RET* mutations have been reported in Project GENIE. Somatic *RET* mutations has been reported in about 25–45% cases of sporadic medullary thyroid carcinoma (MTC), a type of carcinoma with neuroendocrine differentiation. Similarly, *RET* mutations may be associated with the neuroendocrine differentiation in some patients with NECC. In this study, *APC*, which is a negative regulator of the Wnt/β-catenin pathway, was mutated in 3 (12%) patients with NECC. The Wnt/β-catenin pathway plays a key role in the sequential development (initiation, expansion, and transformation) of tumors into cancer [23].

These genetic differences and similarities between NECC and other types of cervical cancer seem to be consistent with the current understanding of NECC; NECC derives from major histological subtypes, such as SCC or ADC, showing trans-differentiation of the neuroendocrine phenotype [24,25]. We hypothesized that these differences could determine the distinct aggressive behavior of NECC. However, survival analysis did not show any prognostic significance of gene mutations limited to NECC, although *KRAS* mutation appears to be a poor prognostic factor. Interestingly, Lyons et al. reported that a mitogen-activated protein kinase 1 inhibitor (trametinib) achieved complete radiologic response in a patient with recurrent NECC with *KRAS* mutation [7].

We also evaluated NECC for actionable mutations with OncoKB evidence levels of 1 to 3B. Actionable mutations were detected in all histological types of cervical cancer, including 11 patients with NECC (44%). *PIK3CA* mutations are the most frequent actionable mutations in cervical cancers and have been approved by the US Food and Drug Administration (FDA) as a predictive biomarker for the use of the PI3K inhibitor, alpelisib [26]. Various PI3K inhibitors have been developed, and numerous clinical trials have been designed to evaluate various solid malignancies, including neuroendocrine tumors [27]. One study demonstrated that combination therapy of etoposide and cisplatin with the PI3K inhibitor, dactolisib, resulted in enhanced cell cytotoxic responses in a NECC cell line by reducing cell viability and increasing cell apoptosis, which may be a potential new treatment strategy against NECC [28]. Four oncogenic *RET* mutations, R886Q, S891L, A883T, and C611Y, were first identified in four patients with NECC. Patients with oncogenic *RET* mutations reportedly benefit from treatment with RET inhibitors, BLU-667 and LOXO-292 [29,30]. Loss-of-function mutations in *RB1* were observed in three patients with NECC in Project GENIE. Although loss of RB1 is not registered in OncoKB, two recent studies described that the deregulation of cell cycle transitions upon loss of RB1 can represent a high dependency on aurora kinases, which can be targeted therapeutically [31,32]. Recently, Ramez et al. reported that 73% of patients with NECC had potentially actionable genetic alterations by analyzing 97 cases of NECC using comprehensive genomic profiling (182–315 genes) [11]. Despite extensive research, some genetic alterations, including *RET* mutations, were first identified in our study. Therefore, the data seem to be insufficient to elucidate the true nature of NECC and, thus, the published data must be integrated and utilized to identify a novel therapeutic target in future studies.

We first provided both TPS and CPS for PD-L1 expression in patients with NECC. Notably, 56% of patients with NECC showed more than one CPS, whereas no cases presented more than 1% TPS. The expression of PD-L1 on tumor and/or immune cells is predictive of the benefit of anti–PD-1 and anti–PD-L1 therapy in several tumor types [33]. Although the expression of PD-L1 in NECC has been consistently reported to be low [3], an exceptional response to nivolumab in recurrent NECC has also been reported [8]. Recently, CPS was used interchangeably with TPS and may be more sensitive than TPS at lower cut-offs in head and neck SCC [34]. Moreover, in one study, it was demonstrated that the expression of PD-L1 on histiocytes in patients with ovarian cancer and melanoma correlated with the efficacy of treatment with either anti-PD-1 alone or in combination with anti-CTLA-4 [35]. The drug, bintrafusp alfa (M7824), was designed to simultaneously bind to two target proteins, PD-L1 and TGF-β, which help prevent the immune system from effectively attacking tumor cells [36]. Bintrafusp alfa shrank tumors in approximately 40% of patients with advanced HPV-associated cancers, including cervical and anal cancers, and showed a manageable safety profile in a phase 1 clinical trial [37]. Overall, the majority of patients with NECC may benefit from precision medicine based on the molecular profiling of genetic alterations and targets of immunotherapy.

Our results show that all patients with NECC were positive for high-risk HPV and 76% of them were positive for HPV-18; however, no significant correlation between HPV genotypes and specific gene mutations was observed. Although HPV-positivity was observed in more than 90% of all cervical cancer patients, NECC was significantly more associated with HPV-18 than other histological types. A large proportion of NECC cases are reportedly caused by high-risk HPV, primarily HPV-16 and HPV-18 [14]. A previous study also reported a predominance of HPV-18 (41% of NECC vs. 10% of other histological types, *p* < 0.001) [38]. Furthermore, HPV-18-related cervical cancers are characterized by marked lymphatic infiltration and poor prognosis [39]. Although the molecular basis is still unknown, several previous studies have described HPV-16 as a favorable prognostic factor, whereas HPV-18 is a poor prognostic factor in HPV-associated cancers, such as cervical cancer and head and neck cancer [39,40,41]. Importantly, considering the close relationship between NECC and high-risk HPV, prophylactic HPV vaccines would be the most effective measure to prevent death from this highly aggressive cancer. Furthermore, immunotherapy or therapeutic vaccines may be developed as an option for HPV-associated cancer.

Frequent HPV-positivity in NECC also has diagnostic utility for detection of a primary site of metastatic NEC. A significant subset of patients with advanced neuroendocrine carcinomas (NECs) are diagnosed after the detection of distant metastases. However, 11–22% of NECs are reportedly of unknown origin [42]. Considering that HPV-positive NECs originate from HPV-associated cancers including cervical, anal, and oropharyngeal cancer [43], clinicians should consider NECs of these primary sites as differential diagnosis when they encounter HPV-positive NEC of unknown primary origin. In addition, frequent HPV-positivity in NECC would also be useful for distinguishing between primary small cell lung carcinoma, which is HPV-independent [44], and lung metastasis of NECC.

This study has several limitations that need to be addressed. First, the number of patients with NECC included in this study was limited. The survival analysis was not robust due to the small number of cases, and further external validation should be performed in a future study. However, considering its rarity, the mutational analysis of 25 cases of NECC has been a relatively large cohort in Asia. Second, targeted sequencing for mutational analysis used in this study contained only hotspot mutations in 50 cancer-related genes. Nevertheless, we could identify new therapeutic targets for NECC, such as alterations in *RET*, even for limited genes and their regions. Plans are currently underway to test therapies targeting the candidate identified in this study.

## 5. Conclusions

We demonstrate genetic similarities and differences between NECC and other histological types of cervical cancers. *PIK3CA* (32.0%), *TP53* (24.0%), *SMAD4* (20%), *PTEN* (20.0%), *STK11* (16.0%), and *RET* (16.0%) were frequently altered in patients with NECC. Nineteen patients (76%) with NECC could potentially benefit from molecular targeted therapies and immune checkpoint inhibitors. Furthermore, high-risk HPV infections, particularly HPV-18, may play a critical role in the carcinogenesis and aggressive behavior of NECC. Therefore, the current HPV vaccines could prevent a large proportion of NECC cases. Collectively, this study reveals specific gene mutation profiles and HPV status in NECC, which can provide a basis for the development of novel therapeutic strategies for this highly aggressive malignancy.

## Figures and Tables

**Figure 1 cancers-13-01215-f001:**
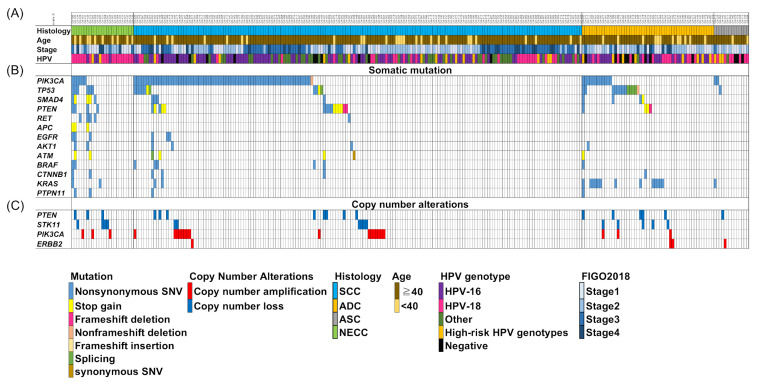
Somatic alterations in cervical cancer and associated clinicopathological features. Two hundred and seventy-two cases were categorized according to their (**A**) histological types and clinicopathological features, (**B**) major mutated genes of neuroendocrine carcinoma, and (**C**) copy number alterations. Mutated genes are color-coded according to their mutation type.

**Figure 2 cancers-13-01215-f002:**
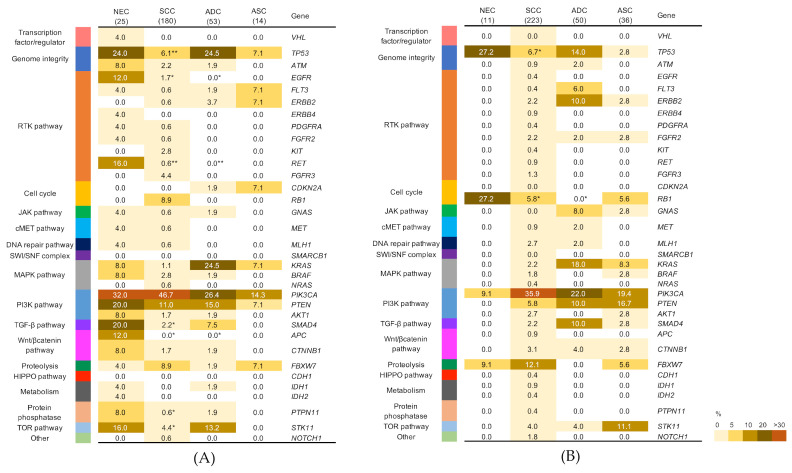
Association between genetic alterations and histological types in cervical cancer. Percentages of samples mutated in individual tumor types are shown. (**A**) the present study, (**B**) Project GENIE v8.0. The *p*-value was calculated using the Fisher’s exact test, * *p* < 0.05, ** *p* < 0.001. NEC: neuroendocrine carcinoma; SCC: squamous cell carcinoma; ADC: adenocarcinoma; ASC: adenosquamous carcinoma.

**Figure 3 cancers-13-01215-f003:**
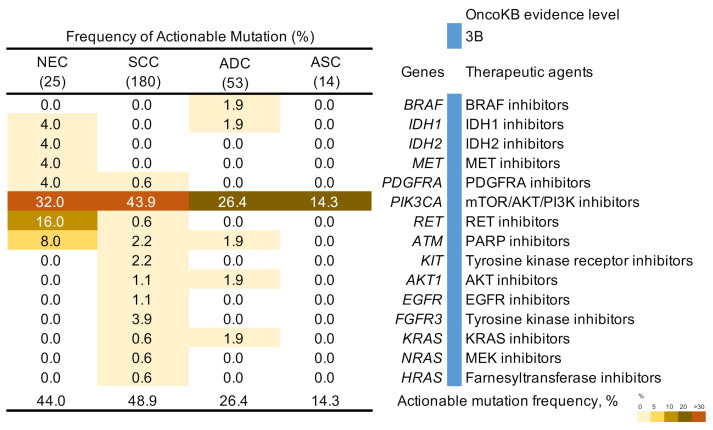
Frequency of actionable genetic mutations in cervical cancers. Percentages of samples mutated in individual tumor types are shown. NEC, neuroendocrine carcinoma; SCC, squamous cell carcinoma; ADC, adenocarcinoma; ASC, adenosquamous carcinoma.

**Figure 4 cancers-13-01215-f004:**
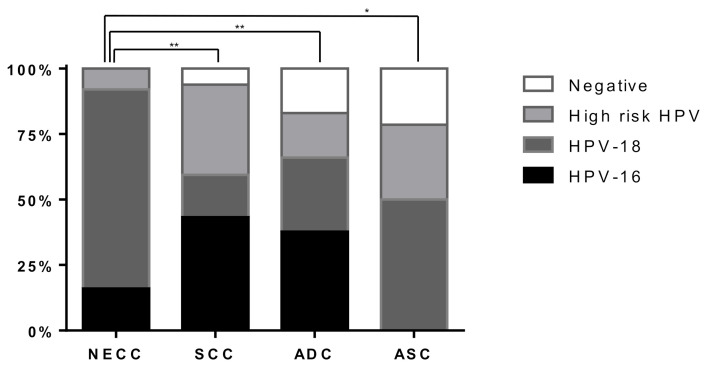
Correlation between histological types and human papillomavirus (HPV) genotypes in 272 cervical cancer specimens. HPV genotypes were compared with neuroendocrine carcinoma (NEC) and other histological types of cervical cancer, including squamous cell carcinoma (SCC), adenocarcinoma (ADC), and adenosquamous carcinoma (ASC). The P-value was calculated using the Fisher’s exact test, * *p* < 0.05, ** *p* < 0.01.

**Table 1 cancers-13-01215-t001:** Clinicopathological characteristics of 272 patients with cervical cancers.

Characteristics		NECC (25)	SCC (180)	ADC (53)	ASC (14)
Age (Year)	Median (Range)	43 (2868)	55 (25–89)	51 (30–82)	47 (37–60)
Stage (FIGO2018), n (%)	I	10 (40.0)	40 (22.2)	18 (34.0)	7 (50.0)
	II	7 (28.0)	58 (32.2)	25 (47.2)	6 (42.9)
	III	4 (16.0)	60 (33.3)	3 (5.7)	1 (7.1)
	IV	4 (16.0)	22 (12.2)	7 (13.2)	0 (0.0)
HPV positivity, n(%)		25 (100)	169 (93.8)	44 (83.0)	11 (78.6)
Treatment, n(%)	Surgery only	6 (24.0)	34 (18.9)	23 (43.4)	4 (28.6)
	RH	6 (24.0)	33 (13.3)	21 (39.6)	4 (28.6)
	RH+PAN	0 (0.0)	1 (0.6)	1 (1.9)	0 (0.0)
	TAH+BSO+PLND+OMT	0 (0.0)	0 (0.0)	1 (1.9)	0 (0.0)
	Surgery+adj-Treatment	9 (36.0)	57 (31.7)	21 (39.6)	9 (64.3)
	RH	7 (24.0)	43 (20.6)	16 (30.2)	6 (42.9)
	RH+PAN	1 (4.0)	10 (5.6)	3 (5.7)	3 (21.4)
	MRHx+BSO+PLND	0 (0.0)	1 (0.6)	1 (1.9)	0 (0.0)
	TAH+BSO+PLND	0 (0.0)	2 (0.6)	1 (0.0)	0 (0.0)
	TAH+BSO	1 (4.0)	0 (0.0)	0 (0.0)	0 (0.0)
	NACT+RH+adjCT	1 (4.0)	1 (0.6)	0 (0.0)	0 (0.0)
	NACRT+RT	0 (0.0)	1 (0.6)	0 (0.0)	0 (0.0)
	RT	1 (4.0)	25 (13.9)	3 (5.7)	0 (0.0)
	CCRT only	3 (12.0)	62 (34.4)	6 (11.3)	1 (7.1)
	RT following CT	1 (4.0)	0 (0.0)	0 (0.0)	0 (0.0)
	CT only	2 (7.0)	0 (0.0)	0 (0.0)	0 (0.0)
	Palliative care only	2 (7.0)	0 (0.0)	0 (0.0)	0 (0.0)

Abbreviations: NECC, Neuroendocrine carcinoma of the cervix; SCC, Squamous cell carcinoma; ADC, Adenocarcinoma; ASC, Adenosquamous carcinoma; HPV, human papillomavirus; NACT, Neoadjuvant chemotherapy; adj, Adjuvant*;* CT, Chemotherapy; RT, Radiotherapy; CCRT, concurrent chemoradiation therapy*;* NACRT, Neoadjuvant chemoradiation therapy*;* RH, radical hysterectomy; BSO, bilateral salpingo-oophorectomy; PLND, pelvic lymph mode dissection*;* PAN, para-aortic lymphadectomy; TAH, total hysterectomy; OMT, omentectomy*;* MRHx, modified radical hysterectomy.

## Data Availability

The data presented in this study are available in the present manuscript or in the Appendix A.

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
