# Peer review of "Comparative Analysis of Genetic Alterations, HPV-Status, and PD-L1 Expression in Neuroendocrine Carcinomas of the Cervix"

_cancers, 2021, doi:10.3390/cancers13061215_

Round 1

Reviewer 1 Report

Takayanagi et al. evaluated the genetic characteristics of neuroendocrine carcinoma of the cervix (NECC) with cervical cancer of other histological types using targeted sequencing and analysis of data available in public database.

The study is conceptually well-structured and conclusions are both clear and adequately supported by results.

Several comments are suggested to improve on the submitted manuscript:

  • The title of this paper is “Comparative Analysis of Genetic Alterations in Neuroendocrine Carcinomas of the Cervix”. The authors analyzed the genetic profile using the Ion AmpliSeq Cancer Hotspot Panel v2 in NECC. However, they also evaluated other parameters not related to the genetic characterization of this tumor: HPV-positivity and PD-L1 expression. Authors should better contextualize these aspects.
  • A significant subset of patients with advanced neuroendocrine tumors are diagnosed after the detection of distant metastases, and the tissue of origin of the primary tumor cannot be determined in a 11-22% of these neoplasms. Therefore, a detailed molecular and genetic characterization of these tumors is crucial also to improve the diagnostic approach in neuroendocrine tumors. Please structure a short discussion on this point (see and cite: PMID 30991795).
  • The authors should better develop the future perspectives of this study.

Author Response

General response: Thank you for your very positive assessment. We have addressed all of the comments, and hope that our explanations and revisions are acceptable to you. As indicated in the responses below, the manuscript has been improved on account of your suggestions.

Point 1: The title of this paper is “Comparative Analysis of Genetic Alterations in Neuroendocrine Carcinomas of the Cervix”. The authors analyzed the genetic profile using the Ion AmpliSeq Cancer Hotspot Panel v2 in NECC. However, they also evaluated other parameters not related to the genetic characterization of this tumor: HPV-positivity and PD-L1 expression. Authors should better contextualize these aspects.

Response 1: We appreciate this important comment raised by the Reviewer. Accordingly, we have changed the title to “Comparative Analysis of Genetic Alterations, HPV-Status, and PD-L1 Expression in Neuroendocrine Carcinomas of the Cervix”.

Point 2: A significant subset of patients with advanced neuroendocrine tumors are diagnosed after the detection of distant metastases, and the tissue of origin of the primary tumor cannot be determined in a 11-22% of these neoplasms. Therefore, a detailed molecular and genetic characterization of these tumors is crucial also to improve the diagnostic approach in neuroendocrine tumors. Please structure a short discussion on this point (see and cite: PMID 30991795).

Response 2: We appreciate this valuable and critical comment raised by the Reviewer. We completely agree with you. We describe this issue in the Discussion as follows: “Frequent HPV-positivity in NECC also has diagnostic utility for detection of a primary site of metastatic NEC. A significant subset of patients with advanced neuroendocrine carcinomas (NECs) are diagnosed after the detection of distant metastases. However, 11-22% of NECs are reportedly of unknown origin [PMID: 30991795]. Considering that HPV-positive NECs originate from HPV-associated cancers including cervical, anal, and oropharyngeal cancer [PMID: 21997688], clinicians should consider NECs of these primary sites as differential diagnosis when they encounter HPV-positive NEC of unknown primary origin. In addition, frequent HPV-positivity in NECC would also be useful for distinguishing between primary small cell lung carcinoma, which is HPV-independent [PMID: 17727473.], and lung metastasis of NECC.” (Page 11, paragraph 2, line 423-431, in Discussion)

Point 3: The authors should better develop the future perspectives of this study.

Response 3: Thank you for encouraging assessment. We are planning to test therapies targeting the candidate genes in a future study. We are grateful for the time and energy you have expended in making these observations and recommendations.

Reviewer 2 Report

Good sound work. 

I hope authors will test therapies targeting the candidate genes in a future study.

Author Response

Response : Thank you for your encouraging assessment. We are planning to test therapies targeting the candidate genes in a future study. We are grateful for the time and energy you have expended on our manuscript.

Reviewer 3 Report

Yes this is  interesting and  important and filling a  gap.  The area  is  still poorly researched and this relatively large series  is important. There are some similarities with other  high grade  G3NEC in lung  but some differences

Might be  useful to contrast with SCC Ovarian cancers where more data is emerging and different  molecular profile

RET also intriguing with new drugs available, i was  not aware of potential in NET/NEC of  cervix

In NEC Lung  the  p53 /RB1 vs STK111 distinction may have some prognostic  and  predictive value, eg poorer response to platinum based regimens  (Derks et al) . This  could again have  huge implications for  treatment  lines

I consider this a valuable  paper adding to our  knowledge and look forward to see more and  expanded data

The STK111/mTOR pathway again raises issues  of potential value of everolimus

Well  done

Author Response

Response : Thank you for your very positive assessment. We would like to get a chance to test therapies targeting the candidate genes in a future study. We are grateful for the time and energy you have expended in making these observations and recommendations.
